# Prognostic risk models for incident hypertension: A PRISMA systematic review and meta-analysis

**Filip Emil Schjerven** [1]*, **Frank Lindseth**[1], **Ingelin Steinsland**[2]

**1** Department of Computer Science, Norwegian University of Science and Technology, Trondheim, Norway,
**2** Department of Mathematical Sciences, Norwegian University of Science and Technology, Trondheim, Norway

* filip.e.schjerven@ntnu.no

## Abstract

### Objective

Our goal was to review the available literature on prognostic risk prediction for incident hypertension, synthesize performance, and provide suggestions for future work on the topic.

### Methods

A systematic search on PUBMED and Web of Science databases was conducted for studies on prognostic risk prediction models for incident hypertension in generally healthy individuals. Study-quality was assessed using the Prediction model Risk of Bias Assessment Tool (PROBAST) checklist. Three-level meta-analyses were used to obtain pooled AUC/C-statistic estimates. Heterogeneity was explored using study and cohort characteristics in meta-regressions.

### Results

From 5090 hits, we found 53 eligible studies, and included 47 in meta-analyses. Only four studies were assessed to have results with low risk of bias. Few models had been externally validated, with only the Framingham risk model validated more than thrice.

The pooled AUC/C-statistics were 0.82 (0.77–0.86) for machine learning models and 0.78 (0.76–0.80) for traditional models, with high heterogeneity in both groups ($I^2$ > 99%). Intra-class correlations within studies were 60% and 90%, respectively. Follow-up time (P = 0.0405) was significant for ML models and age (P = 0.0271) for traditional models in explaining heterogeneity. Validations of the Framingham risk model had high heterogeneity ($I^2$ > 99%).

### Conclusion

Overall, the quality of included studies was assessed as poor. AUC/C-statistic were mostly acceptable or good, and higher for ML models than traditional models. High heterogeneity

**Data Availability Statement:** All relevant data are within the paper and its Supporting Information files.

**Funding:** The author(s) received no specific funding for this work.

**Competing interests:** The authors have declared that no competing interests exist.

implies large variability in the performance of new risk models. Further, large heterogeneity in validations of the Framingham risk model indicate variability in model performance on new populations.

To enable researchers to assess hypertension risk models, we encourage adherence to existing guidelines for reporting and developing risk models, specifically reporting appropriate performance measures. Further, we recommend a stronger focus on validation of models by considering reasonable baseline models and performing external validations of existing models. Hence, developed risk models must be made available for external researchers.

## Introduction

Hypertension is considered the number one preventable risk factor of cardiovascular disease (CVD) and all-cause death globally [1]. The number of individuals suffering from hypertension effectively doubled in the period 1990–2019 to an estimated 1.4 billion individuals [1,2]. Despite global mean blood pressure (BP) remaining still or even decreased slightly over the past decades due to more effective tools in managing BP, prevalence of hypertension has increased, especially in low-to middle-income countries [3]. Yet, upwards of 50% of individuals participating in structured screening programs were not aware of their elevated blood pressure, regardless of their income level in the country they were from [2,4]. One explanation for this unawareness is the predominantly asymptomatic nature of hypertension, highlighting the importance of individuals paying attention to their BP [2].

Despite the existence of effective prevention strategies in the form of lifestyle management and drug treatment, prevalence of hypertension has steadily increased [2,3]. The potential of these strategies has motivated research into identifying individuals at an earlier stage, including the development of prognostic risk models. Risk models can be seen as one way of moving towards personalized medicine, as risk estimates can be estimated for an individual based on their unique set of clinical predictors [5].

Many risk models for hypertension in the general population have been developed in recent years. Two earlier reviews provided a narrative synthesis of the available risk models at the time, which were mostly developed using traditional regression-based models [6,7]. Recognizing the popularization and availability of machine learning (ML) methods, a third review [8] expanded upon the two prior by including ML models. ML and traditional models were analyzed separately in meta-analyses, where high heterogeneity was noted in both cases. A quality assessment of the literature indicated generally low risk of bias (ROB) for traditional models, while no assessment was made on machine learning models [8].

In all three reviews, little distinction was made between diagnostic risk models and prognostic risk models. Although modelling of diagnostic and prognostic models has similarities and even share the same guidelines for appropriate development, they are different in their aim and intended clinical use. A diagnostic model only extends to estimating the risk of existing disease, whereas prognostic models provide risk estimates for a prediction horizon. This implies a difference in the clinical use case between the two model types. Prognostic models may be an auxiliary tool for clinical practitioners. In providing risk estimates of future incidence, they may allow for early intervention and personalized long-term health planning with the goal of preventing incidence [5].

Considering this, we restricted our focus to only considering prognostic risk models for the general population. Using meta-analyses, we synthesize available evidence to quantitatively

summarize the performance of prognostic risk models in the literature [9,10]. To address the issue of studies reporting multiple results, we used a three-level model for pooling results, allowing a more accurate estimation of heterogeneity and its sources. Our aim was 1) systematically review and update existing knowledge on prognostic risk models for incident hypertension among the general population, 2) characterize studies and methodology used, 3) qualitatively assess the literature, and 4) summarize our findings into specific suggestions for improvement of risk modelling for incident hypertension.

## Method

### Retrieving articles and collection of data

A protocol for this review was registered to PROSPERO (ID: CRD42021226152). The Preferred Reporting Items for Systematic Reviews and Meta-Analyses (PRISMA) review form is supplied in S1 Checklist. The search and selection of records, and extraction of data was conducted by one reviewer (F.E.S.) with another reviewer validating each individual step (F.L.). Disagreements were resolved through discussion with a third reviewer (I.S.).

### Databases used

The databases PubMed and Web of Science were used to search for eligible articles. Google Scholar was used for searching for additional hits among references and records citing already included articles. The exact search phrase can be found in S1 Appendix. The final search was conducted March 2023.

### Inclusion and exclusion criteria

Records were eligible if they fulfilled the following criteria:

- Utilized data from a prospective or retrospective cohort,

- The population at baseline consisted of normotensive adults drawn from the general population,

- The primary goal was the development of a model or tool for risk estimation,

- The outcome was prognostic risk of incident primary hypertension as a binary trait,

- The models were evaluated on a dataset and performance measures were reported,

- Written in English.

  Exclusion criteria were:

- Simulation studies,

- Unpublished research studies,

- Studies concerned with any form of secondary hypertension,

- Studies concerned with any other hypertensive diseases, e.g., gestational, ocular, intra-cranial, pulmonary, isolated systolic, isolated diastolic,

- Association studies or studies where the impact of one or a few similar predictors were the primary focus,

- Reviews of the literature

## Selection of studies

After removing duplicates, articles were subsequently screened by title and abstract for relevancy. Articles selected by their title and abstract were then assessed by their full text for eligibility. Works citing, and references of the articles already included were searched for potentially eligible articles not found during the database search. Primary reasons for exclusions after full-text read were detailed following the criteria.

We used citation tracking of included records to assess whether the clinical impact of any risk models had been validated in any subsequent publications.

## Data extraction

For each included study, we collected Population, Intervention, Comparison, Outcome, Timeframe, and Study design (PICOTS) items and information related to bias assessment and meta-analyses.

We also assessed model availability, i.e., whether the developed models could be readily adopted by external researchers or others. A developed model was deemed available if described or presented in text or figure to a sufficient degree for application, provided as a web-tool, or included as downloadable software linked to the publication.

## Risk of bias

The risk of bias within studies was assessed independently by one reviewer (F.S.) using the 'Prediction model Risk of Bias Assessment Tool' (PROBAST) in a two-step process [11]. Initially, a short form version of PROBAST was used to quickly assess studies following a simplified assessment. While the simplified form is less detailed, it has a perfect sensitivity in recognizing articles with high risk of bias [12]. The articles not marked *high risk of bias* during the initial step were then assessed using the full PROBAST form on all reported models. The bias assessment was subsequently validated by a second reviewer (F.L.). Differences in opinion were resolved through discussion with the third reviewer (I.S.).

The initial step using the short form PROBAST was motivated by its time-saving potential, while still ensuring a perfect true positive rate in studies and models marked high risk of bias. Where the original PROBAST form had unspecified numerical criteria, we used thresholds suggested in the short form version [12]. Specifically, these were:

- The sufficient ratio for events-per-variable (EPV) was set to 20 when candidate predictors could be identified, or 40 for final predictors of the model, if not.

- Deletion of data due to missing covariates was unacceptable if more than 5% of included participants was removed, calculated after appropriate exclusions had been applied.

- Doing univariable as opposed to multivariable predictor selection or lack of optimism assessment could be ignored if the EPV was above 25 for candidate predictors, or 50 for final model predictors if candidate predictors were not detailed.

Note, in the case of an external validation of a model, EPV was considered sufficient if the minority outcome had more than 100 events, as described in PROBAST [11].

## Analyses

We used descriptive statistics to summarize our findings. Within each included article, we identified all applications of risk models for incident hypertension and detailed the method

and setup used in model development and validation. Bias in studies was assessed for describing trustworthiness of model results.

Considering earlier reviews on the topic, it was anticipated that the *Area Under the Receiver Operator Curve*, abbreviated AUROC or just AUC, and the *Concordance-statistic*, abbreviated C-statistic, would be the most reported performance measures [6–8]. These statistics are *discriminatory* measures and are equivalent in the binary outcome setting when info on event times are not used [13]. As for calibration measures, the Hosmer-Lemeshow statistic was expected to be the most widely reported [14]. Meta-analyses on reported AUC/C-stat. measures were fitted separately for traditional models and ML models, as done in an earlier review [8]. Calibration was not used in meta-analyses due to incompleteness and variation in reporting.

Further, based on earlier reviews, we assumed some models had been externally validated by independent researchers. We performed meta-analyses for the Framingham risk model to assess the expected performance and heterogeneity in a situation where variation in model development was not relevant. No other model had been externally validated to an extent that would allow a separate analysis.

Meta-analyses and regressions were calculated using the *metafor* package in R [15]. In articles where risk models for hypertension had been developed or externally validated, it is common to report multiple results. This may be to test various aspects of model development or the use of different datasets. In the context of a meta-analysis or regression, this was a problem due to the possible correlation between results that are reported using the same datasets. Meta-analyses and regressions can accommodate this interdependency if the exact covariances or correlations between results are given. However, estimates for within-study covariance or correlation are often not reported in literature [9], meaning another approach must be used in analyses. Naïve inclusion, i.e., assuming zero within-study correlation, of all results would overemphasize the importance of the studies that reported the most results. Aggregating results per study or randomly selecting singular studies has been proposed as an alternative but would imply the loss of statistical information and is not considered ideal [16]. We opted to address this issue by selecting a subset of results found in included studies as well as applying a three level meta-analysis model that can account for some of the described correlation.

We selected the subset of results by the following considerations:

- For nested models we did not include results for the reduced models.

- When several model performance measures for different validation procedures were reported for a new risk model, only one score was included based on how they were calculated, with priority (from high to low): Bootstrapping/cross-validation, then test dataset results, then development data.

- Models with only one or two predictors were excluded unless derived as such during model development.

- Complete information on AUC/C-stat. standard error or the information needed to approximate it was reported.

- Where discrete models like nomograms or risk scores were derived from continuous models, we included only results from the continuous model if available.

These considerations were made per modelling method, gender, and the studies mean/median follow-up time reported in each article.

In using a three-level model in our meta-analyses, we allow heterogeneity to be estimated at two levels as opposed to just one. We grouped results within the studies they were reported,

modelling heterogeneity at the study level as well as the level of individual results. In practice, if results from the same studies were more similar within each study compared to results across studies, most heterogeneity would be estimated at the study level. Although a three-level model is not a perfect reflection of the true correlation structure, it will likely produce a model closer to reality versus a two-level model ignoring within-study correlation [17]. To evaluate the three-level model compared to a two-level model, we performed a likelihood ratio test.

The three-level model does not fully solve the issue of sample error correlations not being reported in studies. Rather, it produces an approximation assuming all covariances between individual results within the same study are equal [16]. To account for this, we set missing sample-error correlations to zero and subsequently apply the ClubSandwich estimator for three-level models. This estimator is robust to slight model misspecification that arises from ignoring sampling errors [17].

We performed a sensitivity analysis to assess the possible impact of ignoring sample-error correlation: We sampled random sample-error correlations for results within each study and ran our meta-analysis, repeating the procedure a total of 1000 times. This provided distributions of relevant parameter estimates such as heterogeneity and mean effect which allowed us to estimate how much relevant parameters could be affected had we known the correct sample-error correlations.

In the third analysis, we applied a three-level meta-analysis to the external validations of the Framingham risk model. In the case of studies reporting multiple validation results, we fixed the intra-class correlation to that estimated by the meta-analysis on the traditional models. For the external validations of the Framingham risk model, we excluded the results which included diabetics in the data as the original model was not developed for individuals with diabetes [18].

Moderators were included in meta-regressions to assess whether they could explain some of the heterogeneity. We included the following moderators: Region (Americas–Europe–Asia), mean/median age of cohort at baseline, median/mean follow-up time of study, number of participants in study, and the incidence-rate in data. Gender (men–women–all) was included for traditional models, but not for ML models due to homogenous distribution. Mean/median blood pressure of cohort at baseline was considered but was not reported sufficiently for inclusion. Each moderator was tested as a single moderator. In addition, one analysis was performed with all moderators included simultaneously. P-values below 0.05 were considered significant. All meta-analyses and meta-regressions were calculated using the REstricted Maximum Likelihood (REML) estimator.

For all analyses, the AUC/C-stat. was transformed using the *logit* transformation. The logit transform of standard deviations were approximated, and if not reported, fully estimated using the equations provided by Debray et al. [9]. Studies with insufficient information on sampling error or other data needed to estimate standard deviations were left out of meta-analyses.

We did not assess publication bias by any statistical tests or funnel plot asymmetry.

Analyses were calculated using the R language (version: 4.3) and the RStudio IDE, with the software libraries *tidyverse* used to handle data and *metafor* to perform meta-analyses and meta-regressions [15,19–21]. Plots and figures were created using *ggplot and ggh4x* [22,23]. All code and data required to reproduce the results and associated figures are provided in S1 and S2 Files.

## Results

### Study details

A PRISMA flow diagram detailing the search process can be seen in Fig 1. From an initial pool of 5090 unique records provided by our search terms, we found 46 eligible records, with seven

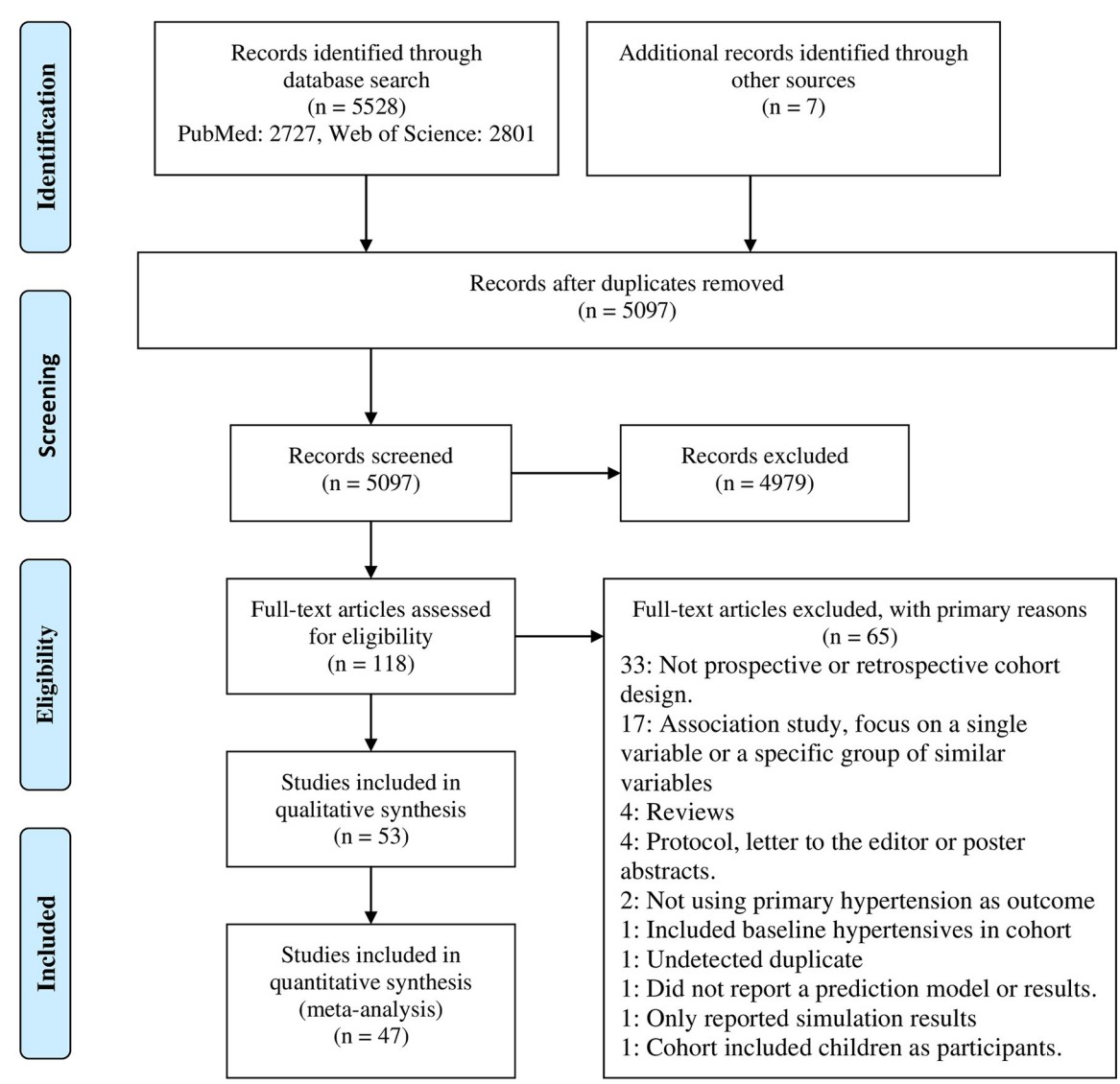

Fig 1. PRISMA diagram of search process and selection of eligible records.

**Table 1. Cohort origin statistics.**

| Cohort origin | Number of studies, n = 53 | Share of total |
|---|---|---|
| China | 12 | 23% |
| USA | 10 | 19% |
| Korea | 8 | 15% |
| Japan | 6 | 11% |
| England | 4 | 8% |
| Iran | 3 | 6% |
| Finland | 2 | 4% |
| Sweden | 2 | 4% |
| Canada | 2 | 4% |
| Brazil | 1 | 2% |
| India | 1 | 2% |
| Germany* | 1 | 2% |
| Denmark* | | |
| Singapore | 1 | 2% |

'*': Cohorts from both countries were used in the same study.

more discovered in citation analysis. In total, 53 articles were included in the review [18,24–75]. The cohort origin of studies is detailed in Table 1 with selected summary statistics presented in Table 2. We did not find any impact studies of the included risk models.

Among the included articles, cohorts from Asia were utilized the most, followed by North America and Europe. Only one article used a cohort from South America. None were found for any African or Oceanian populations. The number of individuals included in cohorts had a median of 6454 individuals and ranged from 297 to hospital electronic medical record (EMR) datasets containing more than 823 000 individuals. All included studies had a follow-up period of one year or more, with a median of five years. Fava et al. was an extreme outlier with a median 23-year follow-up [31]. Lastly, the number of model results reported per study varied from 1 to 22, with a median of 5.

The definition of hypertension used in studies was mostly consistent with ESC/ESH guidelines: 39 studies used systolic blood pressure above 140, diastolic blood pressure above 90, or the use of medication related to managing elevated blood pressure to define hypertension [2]. Use of medication or any existing diagnosis of hypertension was often reported by participants themselves. 10 studies relied on existing diagnosis-codes like 'ICD-10' or related in EMRs, or a diagnosis predetermined by medical professionals. In a single study, an annual questionnaire was used to determine the presence of hypertension. Two studies did not provide details on how hypertension was defined in their study. One individual study used systolic blood pressure above 130, diastolic blood pressure above 80, or the use of medication related to managing elevated blood pressure to define a presence of hypertension. This is in line with the recommendations defined by the American College of Cardiology (ACC) and American Heart Association (AHA) [76].

**Table 2. Summarized characteristics of included studies.**

| | Median | Inter-Quartile Range | Min. | Max. |
|---|---|---|---|---|
| Time between baseline and endpoint, years | 5 | 4–7 | 1 | 23 |
| Cohort size | 6454 | 3983–16860 | 297 | 823627 |
| Results reported per study | 5 | 3–8 | 1 | 22 |

## Modelling methods

Of the 53 included studies, 50 developed a new risk model. Traditional algorithms were used in 44 studies, and machine learning in 14 studies. Hence, eight studies developed at least one model using a method from each group of algorithms.

Studies developing traditional models used mostly the same methods: In cohorts with varying follow-up times, this was Weibull or Cox regression in more recent years. Otherwise, logistic regression was mostly used. 12 of the 44 studies presented a risk score table or nomogram derived from a fitted parametric model, primarily with the intention of simplifying the model for clinical use.

In terms of machine learning algorithms, a total of 13 different algorithms had been used to develop new risk models. The most popular were Random Forest, Artificial neural networks, XGBoost, and Support Vector Machine algorithms. An overview of methods with summarized results from each article is given in Table 3.

## Availability of models

Among the 44 studies developing traditional models, 40 studies had made their final model available. However, among the 14 studies applying machine learning, only four studies had made a developed machine learning model available [34,42,45,52]. These four studies described their models graphically. While some complex machine learning models may be ill-suited for presentation via text or figures, we note that no models had been made available through online resources either.

## Performance measures

The most frequent reported performance measure for discrimination was the AUC or the C-statistic. Calibration measures were reported less consistently and with more variation in method, with the Hosmer-Lemeshow statistic being the most popular. Other notable methods were graphical assessment of calibration plots, curves, or distributions, reporting the predicted-observed ratio, or other statistical tests such as the Greenwood-Nam-D'Angelo test.

While no impact studies were found, we found four studies that reported measures for assessing clinical implication of their model. In these, the *net benefit* of the developed model was assessed and compared to alternatives using decision curves [59,63,75,77].

Some articles reported performance measures for comparing models, such as the *Net Reclassification Index*. A few articles included more performance measures of discrimination, like accuracy, sensitivity, Brier score and others. A recurring issue for measures requiring a defined risk threshold for prognosis was that the threshold was not given. Due to a lack of consistent reporting on these measures, these results were not recorded.

## Validation of models

We observed a large degree of variability in the validation during model development: 11 studies (22%) reported a form of cross-validation, bootstrapping, or other repeated sampling procedure in validating its results. A test set was the primary validation method in 23 articles (46%). 16 studies (32%) reported only results from the data the model was derived from. Additionally, four studies used external datasets to perform validation of their results within the development study, shown in Table 4.

## Externally validated models

Only six risk models were found to be externally validated in subsequent publications on new populations. The Framingham risk model was validated the most, by 16 studies with a large

**Table 3. Summarized AUC/C-stat. results of new models in the included studies.**

| First author, year | Method used | AUC/C-stat. Max. [Median] | | |
|---|---|---|---|---|
| | | Development dataset | Testing dataset | Bootstrap / CV |
| Parikh et al., 2008[18] | Weibull reg. | - | - | 0.788 |
| Kivimäki et al., 2009 [24] | Weibull reg. | - | 0.804 | - |
| Paynter et al., 2009[25] | Logistic reg. | 0.714 [0.708] | 0.705 [0.704] | - |
| Kivimäki et al., 2010 [26] | Weibull reg. | - | 0.799 [0.796] | - |
| Kshirsagar et al., 2010[27] | Logistic reg. | 0.800 [0.768] | 0.776 [0.758] | - |
| Muntner et al., 2010[28] | Poisson reg. | 0.773 [0.731] | - | - |
| Bozorgmanesh et al., 2011[29] | Weibull reg. | 0.741 [0.720] | - | - |
| | Risk score sheet | 0.734 [0.730] | - | - |
| Chien et al., 2011[30] | Weibull reg. | 0.741 [0.739] | - | - |
| | Risk score sheet | 0.735 [0.734] | - | - |
| Fava et al., 2013 [31] | Cox reg. | 0.664 [0.663] | - | - |
| Carson et al., 2013[32] | Weibull reg. | 0.710 | - | - |
| Lim et al., 2013[33] | Weibull reg. | 0.780 | 0.791 [0.749] | - |
| | Risk score sheet | - | 0.790 | - |
| Völzke et al., 2013 [34] | Bayes network | 0.780 | 0.790 | - |
| Choi et al., 2014[36] | Logistic reg. | 0.973 [0.906] | - | - |
| Lim et al., 2015[37] | Logistic reg. | 0.811 [0.811] | - | - |
| Lu et al., 2015[38] | Logistic reg. | 0.777 [0.685] | - | - |
| Otsuka et al., 2015[39] | Cox reg. | - | 0.861 [0.861] | - |
| | Risk score sheet | - | 0.858 | - |
| Zhang W et al., 2015[40] | Cox reg. | 0.801 [0.778] | - | 0.8 [0.7775] |
| Chen et al., 2016[41] | Cox reg. | 0.761 [0.757] | - | 0.76 [0.7545] |
| Lee et al., 2016[42] | Cox reg. | 0.729 | 0.724 | - |
| | Bayes network | 0.700 | 0.693 | |
| Lim et al., 2016[43] | Logistic reg. | 0.707 | - | - |
| Niiranen et al., 2016[44] | Logistic reg. | 0.733 [0.732] | - | - |
| Ramezankhani et al., 2016[45] | Decision trees | - | 0.810 [0.770] | - |
| Sathish et al., 2016[46] | Risk score sheet from logistic reg. | 0.802 | - | - |
| Kanegae et al., 2018[47] | Cox reg. | 0.885 [0.885] | - | - |
| Patnaik et al., 2018[48] | SVM | 0.9 [0.866] | - | - |
| Sakr et al., 2018[49] | MLP neural net | - | 0.660 [0.655] | 0.670 |
| | Logit boost | - | 0.730 [0.725] | 0.690 |
| | Locally weighted naïve Bayes | - | 0.690 [0.690] | 0.670 |
| | Random Forest | - | 0.820 [0.815] | 0.930 |
| | Bayes network | - | 0.830 [0.830] | 0.700 |
| | SVM | - | 0.720 [0.720] | 0.710 |
| Ye et al., 2018[50] | XGBoost | 0.917 | 0.870 | - |
| Kadomatsu et al., 2019[51] | Logistic reg. | 0.830 [0.818] | - | 0.830 |
| | Risk score sheet | 0.826 | - | - |
| Xu F et al., 2019[52] | Cox reg. | 0.806 [0.806] | 0.771 [0.765] | 0.779 [0.778] |
| | MLP neural net | 0.809 [0.788] | 0.773 [0.764] | - |
| | Naïve Bayes | 0.796 [0.774] | 0.761 [0.76] | - |
| | CART Decision tree | 0.740 [0.730] | 0.722 [0.71] | - |
| Kanegae et al., 2020[53] | XGBoost | 0.976 | 0.877 | - |
| | Logistic reg. | 0.855 | 0.859 | - |
| | Ensemble | 0.992 | 0.881 | - |

*(Continued)*

**Table 3.** (Continued)

| First author, year | Method used | AUC/C-stat. Max. [Median] | | |
|---|---|---|---|---|
| | | Development dataset | Testing dataset | Bootstrap / CV |
| Nusinovici et al., 2020[54] | SVM | - | 0.780 | - |
| | Random Forest | - | 0.765 | - |
| | MLP neural net | - | 0.775 | - |
| | Logistic reg. | - | 0.770 | - |
| | KNN | - | 0.768 | - |
| | GBM | - | 0.767 | - |
| Poveda et al., 2020[55] | Logistic reg. | 0.776 [0.773] | - | 0.766 [0.764] |
| Syllos et al., 2020[56] | Logistic reg. | - | 0.830 | - |
| Black et al., 2021[57] | Logistic reg. | 0.840 | - | - |
| Castela Forte et al., 2021[58] | Decision rules | - | - | 0.700 |
| Deng et al., 2021[59] | Nomogram from logistic reg. | 0.750 [0.605] | 0.824 [0.597] | - |
| Dritsas et al., 2021[60] | Logistic reg. | - | 0.816 [0.811] | - |
| | Naïve Bayes | - | 0.758 | - |
| | KNN | - | 0.653 | - |
| | Decision tree | - | 0.692 | - |
| | Random Forest | - | 0.798 | - |
| | Stacked Random Forest and Logistic reg. | - | 0.823 | - |
| Fang et al., 2021[61] | KNN | - | 0.946 | - |
| | Light-GBM | - | 0.928 | - |
| | KNN and light-GBM ensemble | - | 0.951 | - |
| | SVM | - | 0.889 | - |
| | Random Forest | - | 0.913 | - |
| | MLP neural net | - | 0.895 | - |
| Koohi et al., 2021[62] | Weibull reg. | 0.820 [0.820] | - | 0.820 |
| Niu et al., 2021[63] | Cox reg. | - | 0.786 [0.786] | - |
| | MLP neural net | - | 0.798 [0.794] | - |
| | Random Forest | - | 0.861 [0.849] | - |
| | GBM | | 0.871 [0.863] | |
| Oishi et al., 2021[64] | Cox reg. | 0.812 | - | 0.804 |
| | Risk score sheet | 0.790 | - | - |
| Vaura et al., 2021[66] | Cox reg. | 0.804 [0.802] | - | - |
| Wang et al., 2020 [67] | Logistic reg. | 0.795 | 0.791 | 0.794 |
| | Risk score sheet | - | 0.784 | - |
| Xu Y et al., 2021[68] | Nomogram from logistic reg. | 0.803 | 0.809 | - |
| Chowdhury et al., 2022[69] | Cox reg. | 0.707 [0.770] | 0.770 | - |
| Datta, 2022[70] | Logistic reg. | 0.920 [0.920] | 0.780 [0.775] | - |
| | XGBoost | 0.960 [0.955] | 0.870 [0.855] | - |
| | Light-GBM | 0.960 [0.950] | 0.860 [0.85] | - |
| | LSTM-NN | 0.980 [0.970] | 0.940 [0.92] | - |
| Jeong et al., 2022[71] | Logistic reg. | 0.630 | - | - |
| | Random Forest | 0.825 | | |
| | XGBoost | 0.828 | | |
| Kawasoe et al., 2021 [72] | Logistic reg. | 0.790 [0.760] | 0.790 [0.770] | - |
| Kurniansyah et al., 2022[73] | Logistic reg. | 0.656 [0.627] | - | - |
| Namgung et al., 2022[74] | Weibull reg. | - | 0.741 [0.740] | - |
| | Risk score sheet | - | 0.729 [0.728] | |

(*Continued*)

**Table 3.** (Continued）

| First author, year | Method used | AUC/C-stat. Max. [Median] | | |
|---|---|---|---|---|
| | | Development dataset | Testing dataset | Bootstrap / CV |
| Zhang X et al., 2022[75] | Nomogram from Cox reg. | 0.764 [0.754] | 0.791 [0.7795] | - |

Method and AUC/C-stat. results are summarized for new models per article, method and type of validation procedure used. Medians are reported for articles with >1 model results per validation procedure. 'Development dataset': Results from dataset used to fit the model. 'Testing dataset': Results from a part of the dataset withheld from model-fitting. 'Bootstrap / CV': Results from cross-validation, bootstrapping, or any resampling procedure.

'GMB': Gradient Boosting Machine, 'KNN': K-Nearest Neighbor, 'MLP': Multi-Layer Perceptron, 'Reg.': Regression, 'SVM': Support Vector Machine, 'XGBoost': eXtreme Gradient Boosting.

regional diversity [18]. The validation AUC/C-stat. ranged from 0.537 to 0.840, while the AUC/C-stat. was reported as 0.788 in the development study. Calibration varied from acceptable in some cohorts to severe mis-calibration in others. The second most validated model was the KoGES model, being validated in three external studies [33]. The remaining examples had only one external validation by independent researchers each, see Table 4.

## Variables

The most used variables in studies were age, systolic blood pressure, diastolic blood pressure, Body Mass Index (BMI), smoking, sex, and presence of hypertension in family. The top eight most used variables in the literature were the same used in the Framingham risk model [18]. Note, in five studies applying ML methods, complete information about the variables used in final models was not presented [45,48,50,53,70]. A summarized view of predictors used in studies can be seen in Fig 2.

**Use of genetic information.** In total, we found 10 studies investigating the efficacy of using detailed genetic information like genetic risk scores to improve risk prediction for incident hypertension [31,34,36–38,44,55,58,63,66,73]. In almost all cases, the resulting models' AUC/C-stat. was only slightly higher if not equal compared to a model without genetic information. Most applied traditional regression-based models which were limited to only capturing linear effects. In one study, the inclusion of a genetic or personal risk score (GRS/PRS) only improved modelling results for the ML models, which indicates a non-linear effect of the GRS [63].

## Quality assessment of included studies

In total, four studies had results assessed as low risk of bias, all applying or validating traditional models. In the initial step using the short form PROBAST 41 studies were assessed to only have results with high risk of bias, see Fig 3. The predominant remarks were improper handling or documentation of missing data in participants, too low events-per-variable (EPV) ratio, or lack of or improper optimism assessment of results. In the subsequent step, five of the remaining 12 studies were assessed as high risk of bias using the full PROBAST form, mainly due to lack of or insufficient calibration assessment, see Fig 4. Applicability was mostly unclear due to the use of cohorts with only middle-aged or older individuals, e.g., all being 40 years of age or more.

## Meta-analyses

Due to the low completeness of reporting and lack of consistency in how calibration was assessed, we opted for only using the AUC/C-stat. performance measures in meta-analyses,

**Table 4. External validations of risk models found in literature.**

| Developed by: First author, Year, Model name, Cohort origin | Validated by: First author, year | External cohort origin | External cohort follow-up, years | AUC/C-stat. | Calibration | Note |
|---|---|---|---|---|---|---|
| Parikh et al., 2008 [18], Framingham model, USA | Kivimäki et al., 2009 [24] | England | 5.6 | 0.803 | HL: 11.5 | |
| | Kivimäki et al., 2010 [26] | England | 5.7 | 0.796 | HL: 11.5 | |
| | Muntner et al., 2010 [28] | USA | 1.6 | 0.788 | HL: P < 0.001 "...under-estimated the risk..." | Same subjects |
| | | | 4.8 | 0.792 | - | |
| | Chien et al., 2011 [30] | China | 6.15 | 0.709 | HL: 7.4 | |
| | Carson et al., 2013 [32] | USA | 5 | 0.840 | HL: 249.4 | |
| | | | | 0.810 | - | Using only 'Age x Diastolic BP' coefficient |
| | Lim et al., 2013[33] | Korea | 4 | 0.789 | HL: 29.74 | |
| | Völzke et al., 2013 [34] | Germany | 5.3 | 0.770 | HL: 11.26 | Different datasets |
| | | Denmark | 5.4 | 0.730 | HL: 203.3 | |
| | Zheng et al., 2013 | China | 2 | 0.537 | HL: 2287.1 | Same subjects |
| | | | 4.8 | 0.610 | HL: 8227.1 | |
| | Lim et al., 2016[43] | Korea | 4 | 0.729 | - | |
| | Kadomatsu et al., 2019[51] | Japan | 5 | 0.829 | HL: 3.36 | |
| | Syllos et al., 2020 [56] | Brazil | 4 | 0.827 | HL: 3.78 | |
| | Wang et al., 2020 [67] | China | 6 | 0.787 | - | |
| | Castela Forte et al., 2021[58] | England | 8.8 | 0.6 | - | |
| | Koohi et al., 2021 [62] | Iran | 3 | 0.810 | - | |
| | Chowdhury et al., 2022[69] | Canada | 5 | 0.729 | - | |
| | Namgung et al., 2022[74] | Korea | 4 | 0.73 | HL: 155.39 | Coefficient-model. |
| | | | | 0.728 | HL: 25.22 | Risk score based. |
| Pearson et al., 1990, John Hopkins model USA [78] | Chien et al., 2011 [30] | China | 6.15 | 0.707 | HL: 16.7 | |
| Kivimäki et al., 2009 [24], Whitehall II risk model, England | Chowdhury et al., 2022[69] | Canada | 5.8 | 0.581 | - | |
| Chien et al., 2011[30], Chinese risk model, China | Chowdhury et al., 2022[69] | Canada | 5.8 | 0.732 | - | |
| Lim et al., 2013[33], KoGES model, Korea | Lim et al., 2016[43] | Korea | 4 | 0.733 | HL: 14.85 | Overlap in article authors. |
| | Chowdhury et al., 2022[69] | Canada | 5.8 | 0.737 | - | |
| | Namgung et al., 2022[74] | Korea | 4 | 0.737 | HL: 209.72 | Coefficient-model |
| | | | | 0.733 | HL: 255.63 | Risk score based |
| Wang et al., 2020[67], Rural Chinese risk model, China | Chowdhury et al., 2022[69] | Canada | 5.8 | 0.735 | - | |

*(Continued)*

**Table 4.** (Continued)

| Developed by: First author, Year, Model name, Cohort origin | Validated by: First author, year | External cohort origin | External cohort follow-up, years | AUC/C-stat. | Calibration | Note |
|---|---|---|---|---|---|---|
| Völzke et al., 2013 [34], Bayes network, Germany | | Denmark | 5.4 | 0.770 | HL: 40.6 | External data used within development article. |
| Kanegae et al., 2018[47], Jichi Genki risk model, Japan | | Japan | 2.4 | 0.846 | GNA: 8.7 | |
| Seo et al., 2021[65], Hypertension Index Model for Women, Korea | | Korea | 2 | 0.777 | - | |
| | | | 6 | 0.792 | - | |
| | | | 10 | 0.797 | - | |
| Seo et al., 2021[65], Hypertension Index Model for Men, Korea | | | 2 | 0.718 | - | |
| | | | 6 | 0.726 | - | |
| | | | 10 | 0.714 | - | |
| Namgung et al., 2022[74], F-CAVAS HTN, Korea | | Korea | 4 | 0.852 | HL: 659 | |
| Namgung et al., 2022[74], F-CAVAS HTN Risk score, Korea | | | 4 | 0.784 | HL: 345 | |
| Namgung et al., 2022[74], CAVAS HTN, Korea | | | 4 | 0.853 | HL: 2082 | |
| Namgung et al., 2022[74], CAVAS HTN Risk score, Korea | | | 4 | 0.787 | HL: 1175 | |

'-': Calibration not reported. 'GNA': Green-Namwood-D'Agostino test statistic, 'HL': Hosmer-Lemeshow test statistic.

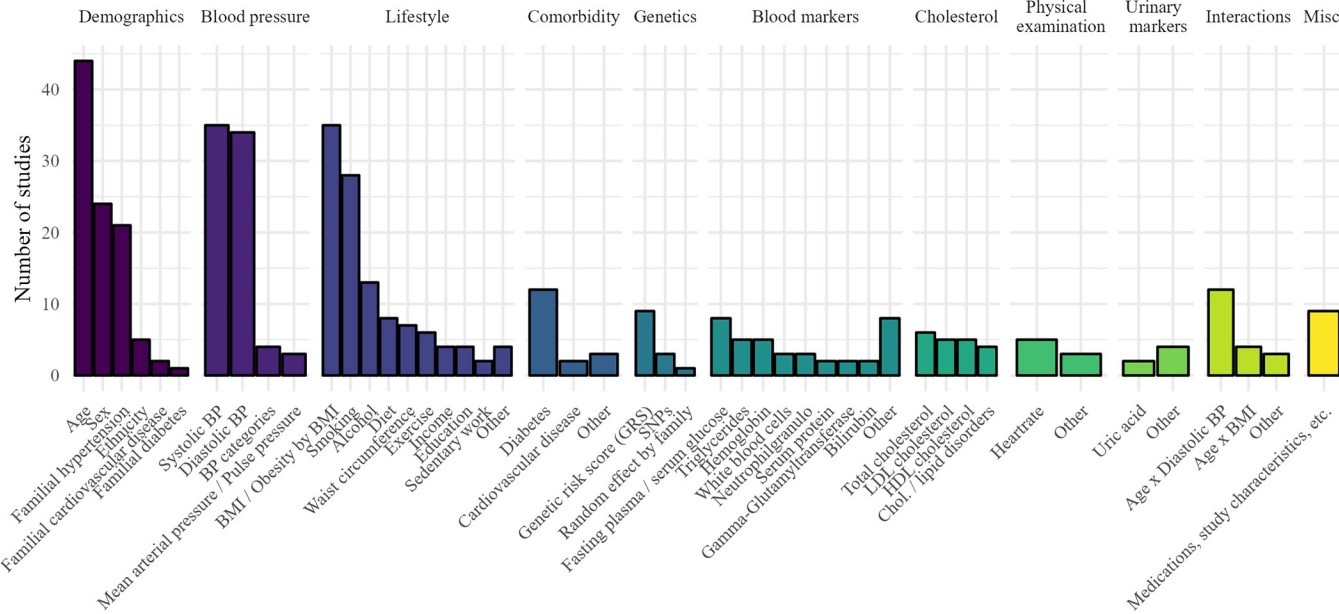

**Fig 2. Variables used in studies.** Variables were counted as those used in any final developed model in a study. We summarize by studies due to variation in the number of developed models per study. Variables used by only one single study were either merged with similar ones, or grouped as"Other" within its category. Note: Variable information from five studies were excluded as they did not report complete information, meaning variable information from 45 studies developing new models are included here. 'BMI': Body Mass Index, 'BP': Blood Pressure,' Chol.': Cholesterol, 'HDL': High-density lipoprotein, 'LDL': Low-density lipoprotein, 'Misc': Miscellaneous, 'SNPs': Single nucleotide polymorphisms.

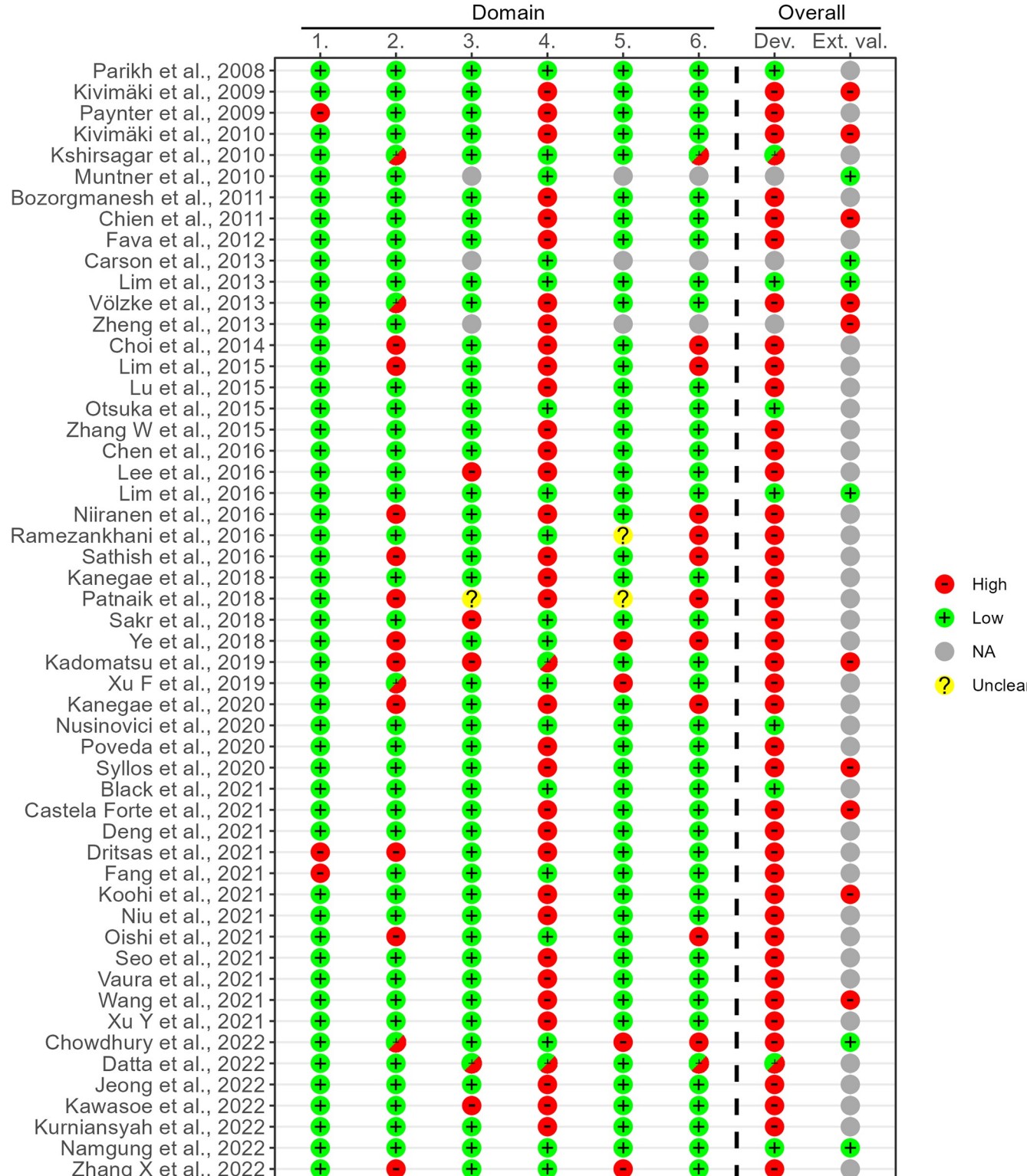

**Fig 3. ROB assessment using short form PROBAST.** PROBAST assessment summarized per study. In short, domains were: 1) Outcome assessment, 2) EPV, 3) Continuous predictors handling, 4) Missing data management, 5) Univariable selection of predictors, and 6) Correction for overfitting/optimism. See Venema et al. [12] for more details. Domains 3, 5 and 6 were not applicable for external validations. Six studies had remarks that were only valid for some of the reported results, e.g., due to the Events Per Variable (EPV) criteria being less strict for external validations or different methods used on some of the developed models. These were marked with a mixed "High/Low" symbol on relevant domain or overall assessment. 'Dev.': Developed models, 'Ext. val.': External validations, 'NA': Not applicable.

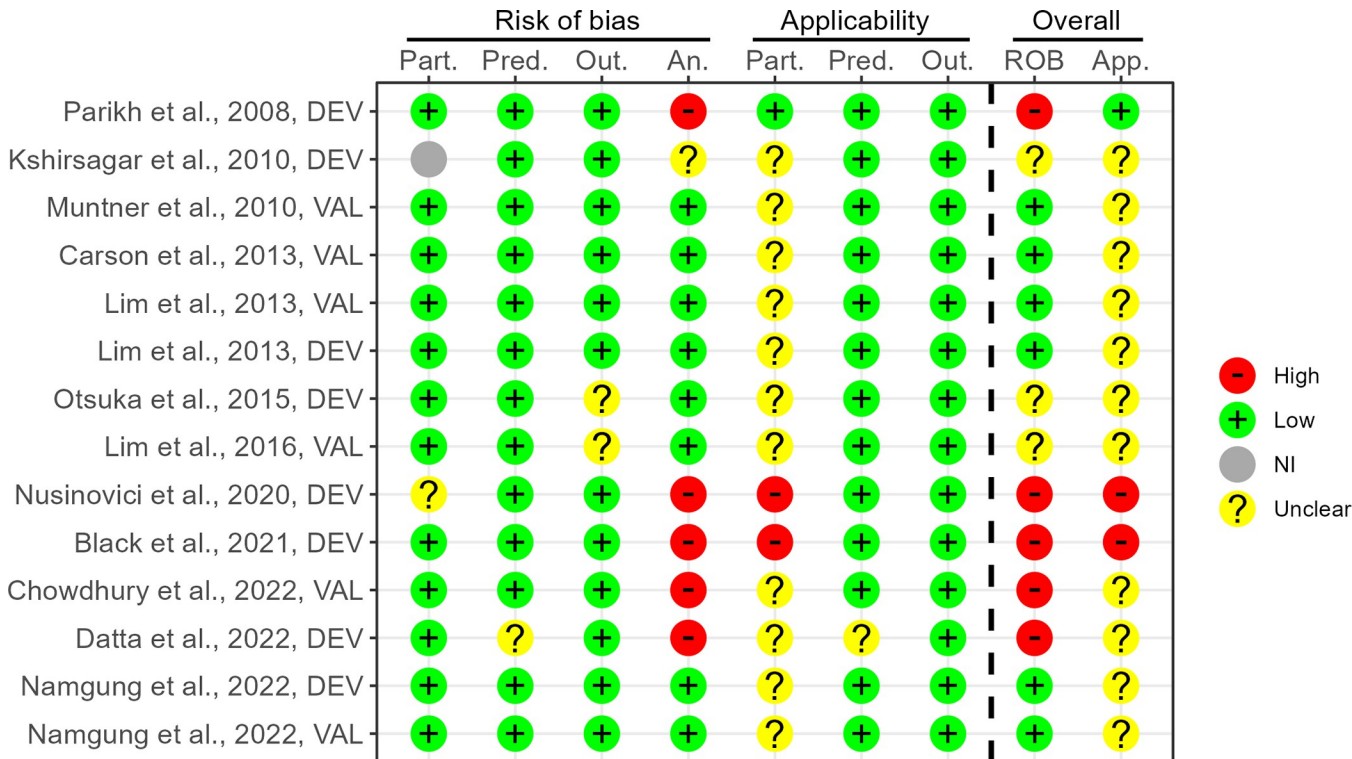

**Fig 4. ROB assessment using long form PROBAST.** PROBAST assessment for studies marked potentially low risk of bias using short form PROBAST. 'An.': Analysis, 'App.': Applicability, 'DEV': Developed models, 'NI': No information, 'Out.': Outcome, 'Part.': Participants, 'Pred.': Predictors, 'ROB': Risk of bias, 'VAL': External validations.

similar to earlier reviews [6–8]. The results of the meta-analyses are summarized in Table 5. Summary characteristics for the model results included in meta-analyses are shown in Table 6.

**Traditional models.** Of the 44 studies developing models using traditional models, 39 had sufficient information for meta-analysis. From these, a total of 46 results were included.

**Table 5. Summarized estimates for AUC/C-stat. and heterogeneity found in meta-analyses.**

| Meta-analyses | Estimated mean (μ), 95% confidence interval (CI), 95% prediction interval (PI) | Heterogeneity estimates | Notes |
|---|---|---|---|
| New developed traditional models. | $\hat{\mu}$ = 0.779<br>CI: (0.762–0.795)<br>PI: (0.660–0.865) | $I^2$: 99.05%<br>*Variability of studies, within*: 9.53%, *between*: 89.52%.<br>*Intra-class correlation*:<br>ICC = 0.904 | 46 results from 39 studies included. |
| New developed ML models. | $\hat{\mu}$ = 0.817<br>CI: (0.767–0.858)<br>PI: (0.547–0.943) | $I^2$: 99.85%<br>*Variability of studies, within*: 39.53%, *between*: 60.32%.<br>*Intra-class correlation*:<br>ICC = 0.604 | 53 results from 13 studies included. |
| External validations of the Framingham risk model | $\hat{\mu}$ = 0.761<br>CI: (0.722–0.795)<br>PI: (0.571–0.883) | $I^2$: 99.45 | 19 results from 16 studies.<br>Using ICC = 0.904 |

'CI': 95% confidence interval, 'ICC': Intra-class correlation, 'PI': 95% prediction interval.

**Table 6. Study characteristics for model results included in the respective meta-analyses.**

| | Median | Inter-quartile range | Min. | Max. |
|---|---|---|---|---|
| Traditional models: | | | | |
| Time between baseline and endpoint, years | 4.75 | 4–6 | 1 | 23 |
| Cohort size | 8319 | 4656–16856 | 297 | 661034 |
| ML models: | | | | |
| Time between baseline and endpoint, years | 5.3 | 2–8.7 | 1 | 10 |
| Cohort size | 8319 | 4592–33289 | 1605 | 680810 |
| Framingham validations: | | | | |
| Time between baseline and endpoint, years | 5 | 4–5.65 | 1.6 | 8.8 |
| Cohort size | 5423 | 3475–13678 | 1605 | 69918 |

Five studies contributed two results and one study contributed three results. Only two of the 39 studies were assessed as having low risk of bias for model development, with another being unclear. Model fit was significantly improved using a three-level model compared to a two-level model (P = 0.0119).

The estimated mean AUC/C-stat. was 0.779 (95% CI: 0.762–0.795). Heterogeneity was high ($I^2$: > 99%, Cochrane's Q = 7325, P < 0.0001). Hence, the 95% prediction interval for new risk models was wide, at 0.660–0.865. The intra-class correlation for each data-source was estimated to be 90.4%, indicating variation was largely due to differences between studies. The contribution of each study to the pooled estimates was calculated to a median of 2.6% and all within 1.98% to 2.83%, meaning that influence of individual studies on our estimates was relatively evenly distributed. A forest plot is provided in Fig 5.

When including moderators, the mean/median age of the cohort at baseline (P = 0.0246), time from baseline to outcome determination (P = 0.0009), and outcome rate (P = 0.001) were found significant in univariable assessment. Only age was found significant (P = 0.0271) when all moderators were included simultaneously, while heterogeneity persisted ($I^2$: 97%).

**Machine-learning models.** Of the 14 studies applying ML to develop risk models, 13 had sufficient information for meta-analysis. From these, a total of 53 results were included. None of the studies applying ML were assessed to have low risk of bias. The three-level model significantly improved model fit compared to a two-level model (P < 0.0001).

The estimated mean performance was 0.817 (95% CI: 0.767–0.858). Heterogeneity was high ($I^2$: > 99%, Cochrane's Q = 22001, P < 0.0001). The 95% prediction interval for new models was 0.547–0.943. The intra-class correlation for each data-source was estimated to be 60.4%, indicating variation due to differences between studies was moderate. The contribution of each study to the pooled estimates was calculated at a median of 8% and all within 5.6% to 8.95%. Hence, the influence of each individual study was relatively even, despite a large variation in the number of results per study included in the analysis.

In meta-regressions, time from baseline to outcome determination was found significant (P = 0.0283) in univariable assessment. Due to the missing entries, only number of participants, time between baseline and outcome determination, and region were included simultaneously, where time between baseline and outcome determination was again significant (P = 0.0405). High heterogeneity persisted ($I^2$: > 99%). A forest plot is provided in Fig 6.

**Sensitivity analysis of ignoring sample-error correlation.** None of the included studies reporting multiple results reported the sample-error correlation. However, the sensitivity analysis of ignoring sample error correlation concluded that the estimates were mostly unaffected. We suggest that this is likely due to the large scale of heterogeneity relative to the possible sample-error *covariance* that was ignored. We refer to S1 Appendix for more details.

| Study | Cohort origin | Method | | Weight (%) | AUC/C-stat., mean (95% CI) |
|---|---|---|---|---|---|
| Jeong et al., 2022 | Korea | Logistic reg. | | 2.7 | 0.63 (0.63 to 0.63) |
| Fava et al., 2012 | Sweden | Cox reg. | | 2.7 | 0.66 (0.65 to 0.67) |
| Castela Forte et al., 2021 | England | Decision rule | | 2.7 | 0.70 (0.69 to 0.70) |
| Paynter et al., 2009 | USA | Logistic reg. | | 2.5 | 0.70 (0.67 to 0.74) |
| Lim et al., 2016 | Korea | Logistic reg. | | 2.7 | 0.71 (0.70 to 0.71) |
| Lee et al., 2016 | Korea | Cox reg. | | 2.7 | 0.72 (0.72 to 0.73) |
| Bozorgmanesh et al., 2011 | Iran | Weibull reg. | | 1.3 | 0.73 (0.71 to 0.75) |
| Bozorgmanesh et al., 2011 | Iran | Weibull reg. | | 1.4 | 0.74 (0.72 to 0.76) |
| Chien et al., 2011 | China | Weibull reg. | | 2.6 | 0.74 (0.72 to 0.76) |
| Namgung et al., 2022 | Korea | Weibull reg. | | 2.5 | 0.74 (0.71 to 0.77) |
| Chen et al., 2016 | China | Cox reg. | | 1.4 | 0.75 (0.74 to 0.76) |
| Chen et al., 2016 | China | Cox reg. | | 1.4 | 0.76 (0.75 to 0.77) |
| Xu F et al., 2019 | China | Cox reg. | | 1.3 | 0.74 (0.71 to 0.76) |
| Xu F et al., 2019 | China | Cox reg. | | 1.4 | 0.78 (0.76 to 0.80) |
| Kshirsagar et al., 2010 | USA | Logistic reg. | | 2.6 | 0.76 (0.74 to 0.78) |
| Poveda et al., 2020 | Sweden | Logistic reg. | | 2.6 | 0.77 (0.75 to 0.79) |
| Nusinovici et al., 2020 | Singapore | Logistic reg. | | 2.4 | 0.77 (0.74 to 0.80) |
| Chowdhury et al., 2022 | Canada | Cox reg. | | 2.4 | 0.77 (0.74 to 0.80) |
| Kawasoe et al., 2022 | Japan | Logistic reg. | | 2.7 | 0.77 (0.76 to 0.78) |
| Deng et al., 2021 | China | Logistic reg.* | | 2.6 | 0.77 (0.75 to 0.79) |
| Datta et al., 2022 | USA | Logistic reg. | | 1.4 | 0.77 (0.76 to 0.78) |
| Datta et al., 2022 | USA | Logistic reg. | | 1.4 | 0.78 (0.77 to 0.79) |
| Lu et al., 2015 | China | Logistic reg. | | 2.7 | 0.78 (0.77 to 0.79) |
| Zhang W et al., 2015 | China | Cox reg. | | 1.4 | 0.75 (0.75 to 0.76) |
| Zhang W et al., 2015 | China | Cox reg. | | 1.4 | 0.80 (0.79 to 0.81) |
| Niu et al., 2021 | China | Cox reg. | | 2.6 | 0.79 (0.76 to 0.81) |
| Parikh et al., 2008 | USA | Weibull reg. | | 2.5 | 0.79 (0.76 to 0.81) |
| Lim et al., 2013 | Korea | Weibull reg. | | 2.5 | 0.79 (0.76 to 0.82) |
| Zhang X et al., 2022 | China | Cox reg.* | | 2.5 | 0.79 (0.76 to 0.82) |
| Wang et al., 2021 | China | Logistic reg. | | 2.6 | 0.79 (0.78 to 0.81) |
| Kivimäki et al., 2010 | England | Weibull reg. | | 2.5 | 0.80 (0.78 to 0.82) |
| Sathish et al., 2016 | India | Logistic reg.* | | 2.0 | 0.80 (0.74 to 0.85) |
| Vaura et al., 2021 | Finland | Cox reg. | | 0.9 | 0.80 (0.78 to 0.82) |
| Vaura et al., 2021 | Finland | Cox reg. | | 0.9 | 0.80 (0.79 to 0.82) |
| Vaura et al., 2021 | Finland | Cox reg. | | 0.9 | 0.80 (0.79 to 0.82) |
| Kivimäki et al., 2009 | England | Weibull reg. | | 2.6 | 0.80 (0.79 to 0.82) |
| Oishi et al., 2021 | Japan | Cox reg. | | 2.4 | 0.80 (0.77 to 0.83) |
| Xu Y et al., 2021 | China | Logistic reg.* | | 2.5 | 0.81 (0.79 to 0.83) |
| Lim et al., 2015 | Korea | Logistic reg. | | 2.6 | 0.81 (0.80 to 0.82) |
| Dritsas et al., 2021 | England | Logistic reg. | | 2.1 | 0.82 (0.77 to 0.86) |
| Koohi et al., 2021 | Iran | Weibull reg. | | 2.5 | 0.82 (0.80 to 0.84) |
| Kadomatsu et al., 2019 | Japan | Logistic reg. | | 2.2 | 0.83 (0.79 to 0.86) |
| Syllos et al., 2020 | Brazil | Logistic reg. | | 2.5 | 0.83 (0.81 to 0.85) |
| Kanegae et al., 2020 | Japan | Logistic reg. | | 2.6 | 0.86 (0.84 to 0.87) |
| Otsuka et al., 2015 | Japan | Cox reg. | | 2.5 | 0.86 (0.84 to 0.88) |
| Kanegae et al., 2018 | Japan | Cox reg. | | 2.4 | 0.88 (0.86 to 0.90) |
| Total (95% CI) | | | | 100.0 | 0.78 (0.76 to 0.80) |

Test for heterogeneity:
$\tau^2=0.08461$; $\chi^2=7324.84$, df=45, P=0.00; $I^2=99\%$

0.5   0.62   0.75   0.88   1

**Fig 5. Forest plot of traditional models.** The 95% prediction interval for new models extending from the summary diamond on the bottom line was calculated as (0.660–0.865). '\*': The result was obtained from a risk score, or nomogram, developed using that method. 'CI': Confidence interval, 'Reg.': Regression.

**External validations of the Framingham risk model.** From the external validations, we selected the 19 external validation results for the Framingham risk model found in 16 different studies. Of these, only three were assessed as low risk of bias, with another assessed as unclear

| Study | Cohort origin | Method | | Weight (%) | AUC/C-stat., mean (95% CI) |
|---|---|---|---|---|---|
| Lee et al., 2016 | Korea | BN | | 5.8 | 0.69 (0.69 to 0.70) |
| Xu F et al., 2019 | China | CART DT | | 1.5 | 0.70 (0.68 to 0.72) |
| Xu F et al., 2019 | China | CART DT | | 1.4 | 0.72 (0.70 to 0.74) |
| Xu F et al., 2019 | China | MLP NN | | 1.4 | 0.76 (0.74 to 0.77) |
| Xu F et al., 2019 | China | NB | | 1.4 | 0.76 (0.74 to 0.78) |
| Xu F et al., 2019 | China | NB | | 1.4 | 0.76 (0.74 to 0.78) |
| Xu F et al., 2019 | China | MLP NN | | 1.4 | 0.77 (0.75 to 0.79) |
| Sakr et al., 2018 | USA | MLP NN | | 1.4 | 0.67 (0.66 to 0.68) |
| Sakr et al., 2018 | USA | LWNB | | 1.4 | 0.67 (0.66 to 0.68) |
| Sakr et al., 2018 | USA | Logit boost | | 1.4 | 0.69 (0.68 to 0.70) |
| Sakr et al., 2018 | USA | BN | | 1.4 | 0.70 (0.69 to 0.71) |
| Sakr et al., 2018 | USA | SVM | | 1.4 | 0.71 (0.70 to 0.72) |
| Sakr et al., 2018 | USA | RF | | 1.4 | 0.93 (0.93 to 0.93) |
| Dritsas et al., 2021 | England | KNN | | 1.8 | 0.65 (0.59 to 0.71) |
| Dritsas et al., 2021 | England | DT | | 1.7 | 0.69 (0.63 to 0.75) |
| Dritsas et al., 2021 | England | NB | | 1.7 | 0.76 (0.70 to 0.80) |
| Dritsas et al., 2021 | England | RF | | 1.4 | 0.80 (0.71 to 0.86) |
| Dritsas et al., 2021 | England | Ridge reg. into RF | | 1.7 | 0.82 (0.77 to 0.86) |
| Ramezankhani et al., 2016 | Iran | QUEST DT | | 1.0 | 0.70 (0.66 to 0.73) |
| Ramezankhani et al., 2016 | Iran | C5.0 DT | | 1.0 | 0.72 (0.68 to 0.75) |
| Ramezankhani et al., 2016 | Iran | QUEST DT | | 1.0 | 0.73 (0.70 to 0.76) |
| Ramezankhani et al., 2016 | Iran | CART DT | | 1.0 | 0.73 (0.69 to 0.77) |
| Ramezankhani et al., 2016 | Iran | C5.0 DT | | 1.0 | 0.77 (0.74 to 0.79) |
| Ramezankhani et al., 2016 | Iran | CART DT | | 1.0 | 0.78 (0.75 to 0.80) |
| Ramezankhani et al., 2016 | Iran | QUEST DT | | 1.0 | 0.79 (0.75 to 0.82) |
| Ramezankhani et al., 2016 | Iran | CART DT | | 1.0 | 0.81 (0.77 to 0.84) |
| Ramezankhani et al., 2016 | Iran | C5.0 DT | | 1.0 | 0.81 (0.77 to 0.84) |
| Nusinovici et al., 2020 | Singapore | RF | | 1.7 | 0.76 (0.73 to 0.80) |
| Nusinovici et al., 2020 | Singapore | GBM | | 1.7 | 0.77 (0.73 to 0.80) |
| Nusinovici et al., 2020 | Singapore | KNN | | 1.7 | 0.77 (0.73 to 0.80) |
| Nusinovici et al., 2020 | Singapore | MLP NN | | 1.7 | 0.78 (0.74 to 0.81) |
| Nusinovici et al., 2020 | Singapore | SVM | | 1.7 | 0.78 (0.75 to 0.81) |
| Völzke et al., 2013 | Germany | BN | | 5.6 | 0.79 (0.75 to 0.83) |
| Jeong et al., 2022 | Korea | RF | | 3.6 | 0.82 (0.82 to 0.83) |
| Jeong et al., 2022 | Korea | XGBoost | | 3.6 | 0.83 (0.83 to 0.83) |
| Niu et al., 2021 | China | MLP NN | | 2.6 | 0.80 (0.78 to 0.82) |
| Niu et al., 2021 | China | RF | | 2.6 | 0.86 (0.84 to 0.88) |
| Niu et al., 2021 | China | GBM | | 2.6 | 0.87 (0.85 to 0.89) |
| Ye et al., 2018 | USA | XGBoost | | 5.8 | 0.87 (0.87 to 0.87) |
| Kanegae et al., 2020 | Japan | XGBoost | | 3.6 | 0.88 (0.86 to 0.89) |
| Kanegae et al., 2020 | Japan | XGBoost + log. reg. + RF | | 3.6 | 0.88 (0.87 to 0.89) |
| Datta et al., 2022 | USA | XGBoost | | 1.4 | 0.84 (0.83 to 0.85) |
| Datta et al., 2022 | USA | light-GBM | | 1.4 | 0.84 (0.83 to 0.85) |
| Datta et al., 2022 | USA | light-GBM | | 1.4 | 0.86 (0.85 to 0.86) |
| Datta et al., 2022 | USA | XGBoost | | 1.4 | 0.87 (0.87 to 0.87) |
| Datta et al., 2022 | USA | LSTM NN | | 1.4 | 0.90 (0.90 to 0.90) |
| Datta et al., 2022 | USA | LSTM NN | | 1.4 | 0.94 (0.94 to 0.94) |
| Fang et al., 2021 | China | SVM | | 1.5 | 0.89 (0.88 to 0.90) |
| Fang et al., 2021 | China | MLP NN | | 1.5 | 0.90 (0.89 to 0.90) |
| Fang et al., 2021 | China | RF | | 1.4 | 0.91 (0.91 to 0.92) |
| Fang et al., 2021 | China | light-GBM | | 1.4 | 0.93 (0.92 to 0.93) |
| Fang et al., 2021 | China | KNN | | 1.4 | 0.95 (0.94 to 0.95) |
| Fang et al., 2021 | China | KNN + light-GBM | | 1.4 | 0.95 (0.94 to 0.96) |
| Total (95% CI) | | | | 100.0 | 0.82 (0.77 to 0.86) |

Test for heterogeneity:
$\tau^2=0.34029$; $\chi^2=22000.89$, df=52, P=0.00; $I^2=100\%$

0.5  0.62  0.75  0.88  1

**Fig 6. Forest plot of ML models.** The 95% prediction interval for new models extending from the summary diamond on the bottom line was calculated as (0.547–0.943). 'Method 1 + Method 2': Ensemble of Method 1 and 2. 'Method 1 into Method 2': Outputs from Method 1 were used as inputs to Method 2. 'BN': Bayes Network, 'CI': Confidence interval, 'DT': Decision tree, 'GBM': Gradient Boosting Machines, 'KNN': K-Nearest Neighbor, 'LSTM NN': Long Short-Term Memory Neural Net, 'LWNB': Locally Weighted Naïve Bayes', 'MLP NN': Multi-Layer Perceptron Neural Net, 'NB': Naïve Bayes, 'Reg.': Regression, 'RF': Random Forest, 'SVM': Support Vector Machines, 'XGBoost': eXtreme Gradient Boosting.

| Study | Cohort origin | Follow up, years | | Weight (%) | AUC/C-stat., mean (95% CI) |
|---|---|---|---|---|---|
| Zheng et al., 2013 | China | 2 | | 3.1 | 0.54 (0.52 to 0.55) |
| Zheng et al., 2013 | China | 4.8 | | 3.2 | 0.61 (0.60 to 0.62) |
| Castela Forte et al., 2021 | England | 8.8 | | 6.0 | 0.60 (0.59 to 0.60) |
| Chien et al., 2011 | China | 6.15 | | 5.9 | 0.71 (0.69 to 0.73) |
| Lim et al., 2016 | Korea | 4 | | 6.0 | 0.73 (0.72 to 0.73) |
| Chowdhury et al., 2022 | Canada | 5.8 | | 5.9 | 0.73 (0.71 to 0.75) |
| Völzke et al., 2013 | Denmark | 5.4 | | 5.9 | 0.73 (0.71 to 0.75) |
| Namgung et al., 2022 | Korea | 4 | | 5.8 | 0.73 (0.70 to 0.76) |
| Völzke et al., 2013 | Germany | 5.3 | | 5.4 | 0.77 (0.72 to 0.81) |
| Wang et al., 2021 | China | 6 | | 6.0 | 0.79 (0.78 to 0.80) |
| Lim et al., 2013 | Korea | 4 | | 5.8 | 0.79 (0.76 to 0.81) |
| Muntner et al., 2010 | USA | 1.6 | | 3.1 | 0.79 (0.77 to 0.80) |
| Muntner et al., 2010 | USA | 4.8 | | 3.1 | 0.79 (0.78 to 0.81) |
| Kivimäki et al., 2010 | England | 5.7 | | 5.8 | 0.80 (0.77 to 0.82) |
| Kivimäki et al., 2009 | England | 5.6 | | 5.9 | 0.80 (0.79 to 0.82) |
| Koohi et al., 2021 | Iran | 3 | | 5.8 | 0.81 (0.79 to 0.83) |
| Syllos et al., 2020 | Brazil | 4 | | 5.8 | 0.83 (0.81 to 0.85) |
| Kadomatsu et al., 2019 | Japan | 5 | | 5.7 | 0.83 (0.80 to 0.85) |
| Carson et al., 2013 | USA | 5 | | 5.9 | 0.84 (0.83 to 0.85) |
| Total (95% CI) | | | | 100.0 | 0.76 (0.72 to 0.80) |

Test for heterogeneity:
$\tau^2$=0.15865; $\chi^2$=4268.19, df=18, P=0.00; $I^2$=99%

X-axis: 0.5  0.62  0.75  0.88  1

**Fig 7. Forest plot for external validations of the Framingham risk model.** The 95% prediction interval extending from the summary diamond was estimated as (0.571–0.883). 'CI': Confidence interval.

risk. Three studies included two results. In Völzke et al. [34], the two results were derived from unrelated cohorts, and we considered these as independent. For the other two, we fixed the intra-class correlation to 0.904 as estimated for the traditional regression-based models and applied a three-level model. We estimated the mean performance to 0.761 (95% CI: 0.722–0.795) with high heterogeneity (I2: > 99%, Cochrane's Q = 4268, P < 0.0001), and subsequently the 95% prediction interval as 0.571–0.883. The cohorts mean/median age at baseline was found significant (P = 0.013) in univariable assessment, without affecting heterogeneity much ($I^2$: > 99%). Simultaneous inclusion of moderators was not performed due to the relatively low number of results available for the meta-regression. A forest plot is provided in Fig 7.

## Discussion

Many risk models for incident hypertension have been developed in recent years with half of the included articles in this review being published in 2018 or later. Concurrent with the substantial increase in the number of relevant articles, a large variation was found in terms of how study cohorts were organized, which variables were used in modelling, and methods used for model development. Notably, while 15 different countries had been represented in study cohorts, only one study used a South American population, and no African or Oceanian populations had been used at all.

The inclusion of genetic information along with clinical information was seen in multiple studies yet displayed little comparative improvement to models without it [31,34,36–38,44,55,58,66]. A single exception was found where ML models improved with the introduction of genetic information, but not the traditional model [63]. This might suggest that non-linear modelling should be considered for capturing the predictive information presented by genetic information. While Völzke et al.[34] considered genetic information in the form of individual single nucleotide polymorphisms (SNPs) in Bayesian Networks, there was no direct comparison versus a model without genetic information. Overall, the included genetic information varied from individual SNPs to full genetic or risk scores for individuals or groups.

We found that only a small proportion of studies were assessed to have low risk of bias results. Improper deletion of individuals with missing data, lack of optimism assessment and improper or missing reporting of relevant performance measures were identified culprits in most articles. Extra care should go into interpreting these reported results, as results may be over-confident, and performance might not be as expected when the model is applied on a new cohort. Notably, most studies assessed as having high risk of bias had similar issues in its study methodology, as identified by the simplified PROBAST form. To improve reporting and study quality, the Transparent reporting of a multivariable prediction model for individual prognosis or diagnosis (TRIPOD) guidelines and the PROBAST assessment criteria themselves may be useful [11,12,79,80].

In the meta-analyses, the pooled effect for ML model was higher than traditional models. However, while the proportion of heterogeneity was similar for both modelling types ($I^2$: > 99%), the scale was far higher for ML models. As such, the 95% prediction interval for new risk models using ML was wider, limiting its usefulness. Traditional models had lower mean effect and a narrower prediction interval, suggesting it as a more conservative approach. However, we note that the scale of heterogeneity is likely to also be affected by our selection of model results. While only 13 of 47 studies included in meta-analyses used machine learning models, we included 53 results from these. This was far more results per study compared to the 46 results from 39 studies using traditional regression-based models.

The few studies applying machine learning may partly explain why only one moderator was found significant in explaining heterogeneity. As moderators largely described study characteristics, the actual number of unique datapoints for these were 13, i.e., the number of studies applying ML. Using the three-level model, the repeated moderator values were accounted for to some extent, as shown by the more even distribution of influence of each study on the pooled estimates.

For studies applying traditional models, time from baseline to outcome determination, outcome rate, and the median/mean age of the cohort at baseline was found significant when included as individual moderators, with only baseline age significant upon including all. Baseline age was also significant as a moderator for the Framingham external validations.

Studies with younger individuals in their cohorts reported better results. Increased age is a known risk factor for developing hypertension, hence it's closely connected to both outcome-rate and follow-up time. Older individuals will have higher outcome-rate, and time from baseline to outcome determination is simply the age-delta of the cohort from baseline to endpoint [2]. Nevertheless, the inclusion of moderators barely explained any heterogeneity in any case, reducing $I^2$ by less than 2%. While other known risk factors of hypertension could be relevant as moderators, e.g., baseline blood pressure, their reporting was inadequate to be tested as moderators without excluding large parts of the included results.

Using the three-level model, analyses estimated considerable correlation within studies for both traditional and ML models. This similarity within studies suggests within-study comparisons using meaningful alternatives are needed for making judgment on the effect of various

modelling choices. Hence, the effect of using a machine learning model, the utility of different data sources, or the inclusion of a new sub-group of individuals can only be assessed meaningfully when compared against alternatives within the same study.

As we meta-analyzed the external validations of the Framingham risk model, we could investigate a case independent of variation in model development methods. Even so, heterogeneity was estimated to account for more than 99% of the variation in results, with baseline age as the only significant moderator. As model development was not relevant, the persistent high heterogeneity suggests that heterogeneity was more related to other aspects, e.g., cohort characteristics or the recording of data. While noting that only a single model was considered, this underlines that confidence intervals of results presented in studies should only be considered relevant within the context of the study it is reported in. To exemplify, the external validation AUC/C-stat. reported for the Framingham risk model ranged from 0.537 to 0.84. This was a far higher variation than suggested by the bootstrapped optimism of 0.0003 or the 95% confidence interval of 0.733 to 0.803 reported in its development article [18].

In 13 of the 16 studies that externally validated a model, a new risk model was presented as well. External validation of a risk model can go beyond simple application of the model in a new population. Several methods may be tested to see if an external model can be made effective in a new population with relatively little effort, e.g., by recalibration or re-estimation of coefficients [81]. The advantage of a thorough external validation of existing models when a new risk model is proposed is two-fold: Failing to obtain favorable results using external models would argue for the creation of a new risk model. Simultaneously, a thorough development of a new risk model will likely produce a best-case scenario performance-wise, which could serve as context for the external validation. Lastly, we note that four of the included studies validated multiple models [30,43,69,74].

External validations are useful for testing performance outside of the development cohort and require that risk models are made available. Only four studies applying machine learning had made one of their final models readily available for external researchers, likely explaining partly why no machine learning models were found to be externally validated by independent researchers. The four ML models that were available had low complexity, allowing a full graphical presentation, e.g., a Bayesian network or a decision tree [34,42,45,52]. No models were found to be shared by online resources. Traditional models are easier to share without resorting to online resources. Several studies specifically emphasized application in clinical practice as a motivation and presented simplified versions of their risk model for easier use by clinicians in the form of nomograms, risk scores or decision rules.

Another aspect that challenges external validation and reproducibility, is the increasing use of datasets derived from Electronic Medical Records (EMR). These often exceed traditional study cohorts in both number of participants and the amount of clinical information, increasing the information load underlying any risk model development. This suggests that reporting should be even more rigorous. As an example of the opposite, in three studies [50,53,70] developing risk models using EHRs, a complete list of variables used in the final models was not reported.

Most included studies only reported discrimination performance measures, especially in studies where ML models were developed. Neglecting other performance measures such as calibration or clinical impact is common, although discrimination ultimately only provides a partial view of a model's total performance [6–8,82,83].

## Limitations of our study

We note that we have included fewer studies than earlier, relevant reviews. While our inclusion criteria were more restrictive, there can be variation in distinguishing prediction model

development from association studies due to similarities in how models are developed, reported, and assessed. We excluded several studies with prognostic models where the focus was fixed on one or a specific set of similar variables as none of these followed recommended procedures for creating risk models. Further, they were often exclusively focused on their specific research niche, implying exponentially more effort would be needed to identify all such studies. Most included studies using genetic information were edge-cases in this sense but were included as they were explicitly labeled as risk scores or risk models for incident hypertension.

Applying the original PROBAST framework upon studies developing ML models may be ill-advised. However, we deemed it relevant as all models were developed for a similar purpose as the traditional models. With the publication of the PROBAST-AI framework, better assessment of risk models based on ML will likely be possible [65].

A significant limiting factor was that we only focused on the performance measure of AUC/C-statistic. Both discrimination and calibration should be assessed simultaneously in meta-analysis to increase power of the analysis [9]. Incomplete reporting as well as variation in methods used meant we were unable to incorporate it into our meta-analyses [14,82].

Lastly, we did not assess publication bias of our included results, similar to an earlier, relevant review [8]. Assessing publication bias was not emphasized in a methodological guideline for systematic reviews on prediction models [9].

## Conclusion

The increase in the number of articles and research-effort relevant for hypertension risk modelling may produce insights on creating better models, highlight limitations of existing ones, and contribute to determine how well risk may be predicted in different populations.

We found 53 studies focused on developing or validating a prognostic risk model for incident hypertension. There was rich diversity in cohort origin, methods applied, and subsequent results obtained. The quality of studies was found to be poor, with only a small minority assessed as low risk of bias using the PROBAST framework. Moreover, specific issues for the studies developing ML models were developed models not being made available and incomplete reporting of the used input variables.

We applied a three-level model meta-regression to analyze the reported AUC/C-statistics, as it was the only performance measure reported to a sufficient degree. Model discrimination was found to be acceptable to good in many cases, and seemingly higher for ML models than traditional models. However, high heterogeneity was seen for both model groups. This suggests considerable variability in the performance of new models.

Only one model, the Framingham risk model, had been externally validated more than three times, and we found large heterogeneity in these external validations. This indicates that there is also large variability in how well models translate to new populations. Despite this, only 16 of 53 included studies reported doing an external validation of an existing model.

Based on our findings, we have identified several items that can enable the research community to better assess hypertension risk models. Broader adherence to existing guidelines for reporting and developing risk models like TRIPOD, and specifically reporting appropriate performance measures beyond discrimination, can help improve the quality of reporting. Further, we recommend a stronger focus on validation so that sources of improvement in risk modelling are identifiable and existing risk models are evaluated. This implies considering reasonable baseline models and performing external validations of existing models. To enable this, any developed risk models and required information for practical use need to be made available for external researchers.

## Supporting information

**S1 Checklist. PRISMA checklist for systematic reviews.**
(DOCX)

**S1 Appendix. Search phrases and sensitivity analysis of meta-analyses.**
(DOCX)

**S1 File. R code used for analyses and graphics.**
(ZIP)

**S2 File. Primary data extracted from studies included in the review.**
(XLSX)

## Author Contributions

**Conceptualization:** Filip Emil Schjerven.

**Data curation:** Filip Emil Schjerven, Frank Lindseth.

**Formal analysis:** Filip Emil Schjerven.

**Investigation:** Filip Emil Schjerven, Frank Lindseth, Ingelin Steinsland.

**Methodology:** Filip Emil Schjerven, Ingelin Steinsland.

**Software:** Filip Emil Schjerven.

**Supervision:** Ingelin Steinsland.

**Validation:** Filip Emil Schjerven, Frank Lindseth.

**Visualization:** Filip Emil Schjerven.

**Writing – original draft:** Filip Emil Schjerven.

**Writing – review & editing:** Filip Emil Schjerven, Frank Lindseth, Ingelin Steinsland.

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
