## [Decision Letter · Decision Letter 0]

18 Sep 2023

PONE-D-23-22153Prognostic risk models for incident hypertension: A PRISMA systematic review and meta-analysis.PLOS ONE

Dear Dr. Schjerven,

Thank you for submitting your manuscript to PLOS ONE. After careful consideration, we feel that it has merit but does not fully meet PLOS ONE’s publication criteria as it currently stands. Therefore, we invite you to submit a revised version of the manuscript that addresses the points raised during the review process.

We look forward to receiving your revised manuscript.

Kind regards,

Amir Hossein Behnoush

Academic Editor

PLOS ONE

3. We are unable to open your Supporting Information file [S1_file.zip]. Please kindly revise as necessary and re-upload.

Reviewers' comments:

Reviewer's Responses to Questions

**Comments to the Author**

1. Is the manuscript technically sound, and do the data support the conclusions?

Reviewer #1: Yes

Reviewer #2: Yes

2. Has the statistical analysis been performed appropriately and rigorously? 

Reviewer #1: Yes

Reviewer #2: Yes

3. Have the authors made all data underlying the findings in their manuscript fully available?

Reviewer #1: Yes

Reviewer #2: Yes

4. Is the manuscript presented in an intelligible fashion and written in standard English?

Reviewer #1: Yes

Reviewer #2: Yes

5. Review Comments to the Author

Reviewer #1: Thank you for the opportunity to review this complex and highly technical paper.

Schjerven and colleagues set out to review available literature (Pubmed and web of science databases) on prognostic risk prediction for incident hypertension, synthesize performance, and provide suggestions for future work on the

topic. The concluded that "Overall, the study quality was assessed as poor. AUC/C-statistic were mostly

acceptable or good, and higher for ML models than traditional models. High

heterogeneity implies large variability in the performance of new risk models. Further,

large heterogeneity in validations of the Framingham risk model indicate variability in

model performance on new populations".

Comments:

Line 40: Which study quality that was assessed as poor? Your study or the studies you considered. Kindly please clarify this.

As machine learning and AI gain traction in health care, the importance of risk models become more apparent. A good risk model should be dynamic, be used in the appropriate patient population, for the right purpose and at the right time. For instance, The Framingham risk model likely indicates variability rightfully so when used in different patient populations from different time periods. Each risk model should therefore be compared on itself and on the patients it was designed for and not with another risk model. With ML, is there a way to compares such risk models to themselves and their appropriate patient populations?

Reviewer #2: Dear Authors,

I have reviewed your manuscript titled "Prognostic risk models for incident hypertension: A PRISMA systematic review and meta-analysis."

It made for a very interesting read. I commend the extensive work put into the analytics of the data available and have no comments/corrections with regards to the technical aspects of your manuscript as this has been well thought out and presented in the manuscript.

I have very, very few grammatical corrections.

1. Line 173: "This correlation is unlikely to find reported in literature." Could this sentence please be revised as it is unclear what it is trying to convey.

2. Line 453: "...was the main culprit" would read better as"...were identified culprits."

I could not agree more to your recommendation of a stronger focus on the validation of risk prediction models as this is a key way to fine-tune these developed risk models and also inform subsequent model development by other groups and in other populations. I might add as well that there is a dire need for hypertension risk prediction models development and validation utilizing data from highly at-risk populations such as Africans. It is quite discouraging that none of the included studies included in your work had African and Oceania subjects.

Thank you.

6. PLOS authors have the option to publish the peer review history of their article (what does this mean?). If published, this will include your full peer review and any attached files.

Reviewer #1: No

Reviewer #2: No

---

## [Author Response · Author response to Decision Letter 0]

24 Oct 2023

October 24, 2023

Amir Hossein Behnoush

Academic Editor

PLOS ONE

Dear Academic Editor Dr. Behnoush,

I hope this message finds you well. I am writing to respond to the reviewers' comments on our manuscript titled " Prognostic risk models for incident hypertension: A PRISMA systematic review." with the reference number PONE-D-23-22153, which we submitted to PLOS ONE. We appreciate the thorough review and valuable feedback provided by the reviewers and the editor.

We would like to thank the reviewers and the editor for their time and effort in assessing our manuscript. Their comments and suggestions have contributed to improving the quality and clarity of our research.

In response to the reviewers' comments, we have revised our submission to address their concerns and improve the manuscript. Below, we provide a summary of the points raised by the reviewers and our corresponding responses and actions taken:

Reviewer #1's Comments:

- Comment 1: Line 40: Which study quality that was assessed as poor? Your study or the studies you considered. Kindly please clarify this.

- Our Response: This sentence referred to the studies included in the review. The specific sentence has been edited to read: “Overall, the quality of included studies was assessed as poor.”

- Comment 2: Each risk model should therefore be compared on itself and on the patients it was designed for and not with another risk model. With ML, is there a way to compares such risk models to themselves and their appropriate patient populations?

 - Our Response: We thank the reviewer for this interesting question. We agree that risk models should primarily be compared against themselves in populations that are like the population it was intended for. Similarly, we believe that there are no foundational differences in how risk models developed using ML should be managed or used compared to those developed using traditional methods. To maximize the scientific value of studies where risk models are presented, researchers should ensure that models are made available and that sufficient information on input variables is given. 

In our review, we found that most studies developing ML models had not made their model available, and multiple cases where only incomplete information on input variables were presented. These issues were non-existent for studies developing traditional models. Researchers developing ML models should put more emphasis on making their produced models easily accessible to ensure scientific value of their work for external researchers. These issues are covered by the sections Availability of models, Variables and in part under Discussion. To further emphasize this finding, we added the following line in the Conclusion: “Moreover, specific issues for the studies developing ML models were developed models not being made available and incomplete reporting of the used input variables.”

Reviewer #2's Comments:

- Comment 1: Line 173: "This correlation is unlikely to find reported in literature." Could this sentence please be revised as it is unclear what it is trying to convey.

 - Our Response: We agree that this sentence was unclear. The specific sentence has been edited and expanded to say: “Meta-analyses and regressions can accommodate this interdependency if the exact covariances or correlations between results are given. However, estimates for within-study covariance or correlation are often not reported in literature [9], meaning another approach must be used in analyses. Naïve inclusion, i.e., assuming zero within-study correlation, of all results would overemphasize the importance of the studies that reported the most results.”

- Comment 2: Line 453: "...was the main culprit" would read better as"...were identified culprits."

 - Our Response: The specific sentence has been edited to say: “…, were identified culprits in most articles.”

1) We have revised the manuscript and file names following PLOS ONE’s style requirements. 

2) Captions for the supporting information files are now supplied at the end of the manuscript. In-text citations have been updated accordingly. 

3) We have successfully tested another version of Supporting Information File 1 [S1_File.zip] on multiple computers and have reuploaded a new file. 

4) The reference list has been thoroughly vetted, and we are now confident in its completeness and correctness. The following changes has been conducted:

One entry has been removed from the reference list. The software package “ggExtra” had been used in creating earlier versions of the figures during drafting but was not used for the final figures included in the article. We noticed that MS Word does not track this removal in ‘Revised Manuscript with Track Changes’, most likely due to an automatic reference manager, Zotero, being used. 

Other changes:

1) Figures 1-7 have been updated to correct minor grammatical errors in labels or axis text.

2) The captions of Figures 2-3 have been updated to improve readability and interpretation of the figures. 

3) Figure 6 has been updated to improve a minor graphical detail: Under the column “Method”, an entry was listed as “Log. Reg. -> RF”. We have changed this to “Log. Reg. into RF” to improve clarity.

4) Abbreviations in figure and table legends have been updated to alphabetical order. 

5) Minor errors in author names, publication year and their subsequent ordering have been corrected in Tables 3 and 4, and Figure 3-7. Ordering has been affirmed as publication year (ascending) followed by alphabetically on first author name for each study. 

We have carefully addressed each comment and made the necessary revisions to the manuscript. Attached, please find the revised manuscript with tracked changes addressing each comment individually.

We believe that these revisions have strengthened the manuscript, and we hope that the reviewers and the editor will find our responses satisfactory. We remain committed to ensuring the highest quality of research and appreciate the opportunity to contribute to PLOS ONE.

Once again, we would like to express our gratitude for the thoughtful review process. We look forward to your feedback and hope for a positive outcome on the publication of our manuscript in PLOS ONE.

Sincerely,

Filip Schjerven

Corresponding Author,

PhD Candidate

Department of Computer Science

Norwegian University of Science and Technology

Filip.e.schjerven@ntnu.no

---

## [Editor Report · Decision Letter 1]

26 Oct 2023

Prognostic risk models for incident hypertension: A PRISMA systematic review and meta-analysis.

PONE-D-23-22153R1

Dear Dr. Schjerven,

We’re pleased to inform you that your manuscript has been judged scientifically suitable for publication and will be formally accepted for publication once it meets all outstanding technical requirements.

Kind regards,

Amir Hossein Behnoush

Academic Editor

PLOS ONE
---

## [Editor Report · Acceptance letter]

18 Dec 2023

PONE-D-23-22153R1 

PLOS ONE

Dear Dr. Schjerven, 

I'm pleased to inform you that your manuscript has been deemed suitable for publication in PLOS ONE. Congratulations! Your manuscript is now being handed over to our production team.

Kind regards, 

on behalf of

Dr. Amir Hossein Behnoush 

Academic Editor

PLOS ONE